# MODEL INFORMATION AS AN ANALYSIS TOOL IN DEEP LEARNING

## ABSTRACT

Information-theoretic perspectives can provide an alternative dimension of analyzing the learning process and complements usual performance metrics. Recently several works proposed methods for quantifying information content in a model (which we refer to as "model information"). We demonstrate using model information as a general analysis tool to gain insight into problems that arise in deep learning. By utilizing model information in different scenarios with different control variables, we are able to adapt model information to analyze fundamental elements of learning, i.e., task, data, model, and algorithm. We provide an example in each domain that model information is used as a tool to provide new solutions to problems or to gain insight into the nature of the particular learning setting. These examples help to illustrate the versatility and potential utility of model information as an analysis tool in deep learning.

## 1 INTRODUCTION

The ultimate goal of many deep learning research has been improving performance on specific datasets, for example, aiming for superior classification accuracy on the ILSVRC challenge. We have witnessed super-human performance on tasks in vision and language processing, but we are still far from understanding how the learning process works and whether they resemble the learning of human. This is partially due to the metric we use providing too little information on the dynamics of learning. Besides, performance on the test set as a sole goal can sometimes even lead to undesirable outcomes or misleading conclusions (Lipton & Steinhardt, 2019).

Recently, several works propose to use the description length of a model (or surrogate estimations thereof) to understand the behavior of learning. In this paper, we refer to such measures of the amount of information content in a model as *model information*. Blier & Ollivier (2018) first demonstrated efficiently encoding a deep neural network with prequential coding technique. Zhang et al. (2020) then proposed an approximation of model information and utilized model information to analyze the information content in a task. They also used model information to explain phenomenons in transfer learning and continual learning. They showed that model information provides a different perspective than performance, directly characterizing the information transfer in the learning process. Voita & Titov (2020) used model information as a probe to analyze what kind of information is present in a text representation model. They claim that model information is more informative and stable than performance metrics when used as a probe.

Model information can provide an informational perspective to learning. It can potentially help to answer questions about learning dynamics, such as how much information exists in a dataset or model, or how much information is transferred in a learning step. Furthermore, we can also reformulate existing problems into problems about information, for example, similarity and capacity, as we will show in this paper. Comparing model information with model performance, model information not only accounts for how good a model can perform but also how fast it learns to perform well (in the sense of sample efficiency, a discussion is given by Yogatama et al. (2019)). The latter can be interpreted as related to the quantity of information transferred in model training.

In this paper, we try to illustrate that model information can provide a framework for analyzing and understanding phenomena in deep learning. In the next section, we provide a general definition of model information, independent of how model information is estimated. We then unify the analysis of fundamental elements of deep learning under the framework of model information. In the

following sections, we use several common problems as examples to show how to use the model information framework as an analysis tool.

## 2    AN INFORMATIONAL PERSPECTIVE TO ELEMENTS OF DEEP LEARNING

To measure the amount of information in a model, Voita & Titov (2020) use the codelength of a dataset subtracting the cross-entropy of the final model:

$$L = L^{\text{preq}}_{\theta_{init}}(y_{1:N}|x_{1:N}) + \sum_i^N \log p_{\theta_N}(y_i|x_i). \tag{1}$$

Zhang et al. (2020) propose to estimate model information by subtracting codelength of $k$ examples (the $k$ examples are independent and different from the training set of size $N$) with an initial model and a final model, and show that it is more reliable than (1):

$$L = L^{\text{preq}}_{\theta_{init}}(y_{1:k}|x_{1:k}) - L^{\text{preq}}_{\theta_N}(y_{1:k}|x_{1:k}). \tag{2}$$

Both methods share an idea that model information can be derived by comparing the codelength of encoding the model and data together, with the codelength of encoding the data alone.

This is essentially an approximation to the Kolmogorov complexity of the "generalizable knowledge" in model $M$ (in Zhang et al. (2020), this is denoted by $K(M) - K(M|T)$). Throughout this paper, we use $D$ to denote a dataset including examples sampled from task $T$, and use $M_D$ to denote a model $M$ that has been trained to converge on $D$. Assume labels $y$ in task $T$ have an underlying generation mechanism $f$: $y_i = f(x_i, \epsilon_i)$ (where $\epsilon_i$ is i.i.d. noise), then the "generalizable knowledge" in $T$ is $K(f)$, and generalizable knowledge in $M_D$ is at most $K(f)$.

We denote the amount of generalizable information a model $M$ contains about a dataset $D$ as $L(M, D)$. Then $L(M_D, D)$ is the "model information": the amount of generalizable information a trained model $M_D$ contains about dataset $D$. Ideally, if learning is "perfectly efficient", then we can expect $L(M_D, D) = K(M_D) - K(M_D|T) = K(f)$.

In this paper, we investigate what can be achieved with a model information measure $L$. We found that model information is a powerful tool for analyzing and understanding many phenomenons in deep learning. This covers the fundamental elements of machine learning: *task*, *data*, *model*, and *algorithm*. In Table 1, we list problems as examples for each domain and summarize how model information can be used to perform analysis of the corresponding problem.

Table 1: Model information as an analysis tool in deep learning

| Element | Relevant problem | Use of model information |
|---|---|---|
| **Task** | Task difficulty | $L(M_D, D)$ |
| **Data** | Domain similarity | $L(M_{D_1}, D_2)$ |
| **Model** (structure) | Model capacity | $\max_D L(M_D, D)$ |
| (parameter) | Ablation study | $L(M_D^{\bar{c}}, D)$ |
| **Algorithm** | Knowledge distillation | $L(M_{D+D_T}, D)$ |

As we shall see, model information can enable attacking some of the above problems from a neat and theoretically-sound perspective. Table 1 is not an exhaustive list but just examples of what one can do with model information. In the following sections, we detail the application of model information to perform analysis of each problem and explain how it can lead to useful insights.

For following experiments in this paper, we use (2) to estimate model information and set the encoding set size $k$=10000, unless otherwise stated.

## 3    TASK: DIFFICULTY

The success of deep learning is marked by its superior performance at solving difficult tasks, like large-scale visual classification and question answering. For evaluating the capability of algorithms,

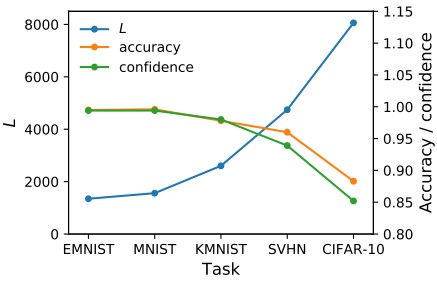

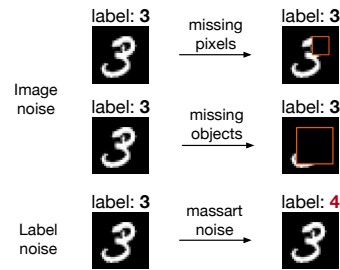

Figure 1: Information content, model performance and model confidence on five tasks.

Figure 2: Injecting three kinds of noise on MNIST.

it is important to understand the difficulty of the tasks and the datasets we have. Human definitions of difficulty are mostly subjective, for example, difficulty scoring, task performance, and time needed to complete the task (Ionescu et al., 2016).

Several metrics have been proposed for measuring the difficulty of a task. Many focus on the complexity of the data. For images, difficulty is linked with image quality, objectiveness, and clutter-ness (Ionescu et al., 2016). For text, the relevant factors include perplexity, reading ease, and diversity (Collins et al., 2018). For general real-valued data, one can measure the distribution of feature value, class separability, and the geometry of class manifolds (Ho & Basu, 2002). There are also other factors that affect the difficulty of a task, for example, class balance and class ambiguity.

As pointed out by Collins et al. (2018), these characteristics all describe some aspect of the task, and difficulty cannot be described by any one of them alone. Complexity measures of input data can sometimes have little impact on task difficulty. An important line of work defines the difficulty of a classification problem by the complexity of the optimal decision boundary (Ho & Basu, 2002). The complexity of the boundary can be characterized by its Kolmogorov complexity or its minimum description length. This coincidences with the idea of the model information. While for simple tasks, one can directly characterize the geometry of the decision boundary, for complex tasks that use a neural network to model the decision boundary, the description length can only be approximated.

With description length, the difficulty of a task can be interpreted as the amount of information needed to solve the task. This amount of information is at most $K(f)$, which is the complexity of the input-output relationship of the task. We measure the model information $L(M_D, D)$ of a trained model, and use it as a measure of the information content in dataset $D$. In Figure 1, we use five 10-class classification tasks to illustrate the idea. The results show that the datasets vary greatly in information content: For classifying digits, SVHN (Netzer et al., 2011) requires more information than MNIST (LeCun et al., 1998) or EMNIST (Cohen et al., 2017), meaning classifying digits in street view is more difficult than in standardized MNIST images. CIFAR-10 (Krizhevsky & Hinton, 2009) is even richer in information content and therefore more difficult, as natural objects are more complex than digits.

There is also a general trend that the more difficult the task, the lower the performance and the confidence of the model. However, performance and confidence are not only determined by task complexity, they are also affected by noise present in the task.

Next we want to disentangle the two factors: task complexity (measured by information) and noise. We apply three kinds of noise on MNIST to represent different forms of noise present in a dataset (Figure 2). There are two kinds of image noise: *missing pixels*, where a small random block of pixels is missing in the image, *missing objects*, where a large portion of the digit is missing, and one form of label noise: *massart noise*, where a small ratio of labels are replaced by random labels. To see how different kinds of noise affect model behavior, we plot model information, model accuracy, and confidence in Figure 3.

What we observe is that information complexity and noise are two independent dimensions that affect performance on a task. Pixels missing from the image will make the task more complex and require more information to correctly classify the digits. This is because one must learn more features corresponding to each digit, as any of the features could be absent at any time. However, the

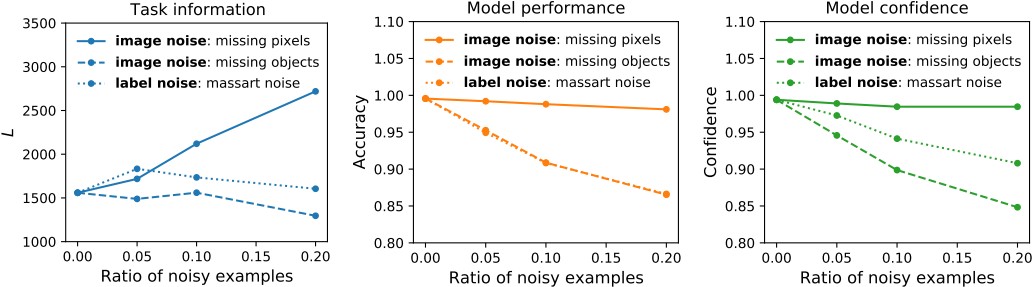

Figure 3: Task information, model performance, and model confidence, with different kinds and varying levels of noise.

model can still perform well after learning enough information about the task. Missing objects and wrong labels, on the other hand, are pure noise that does not affect the information complexity of the task. They effectively make some of the examples uninformative and simply confusing, which makes the model less confident in its predictions.

## 4 DATA: DOMAIN SIMILARITY

Understanding the data is a fundamental element of understanding learning. Labeled data for deep learning comes from many sources and domains, and people seek to train models that can adapt well to a new domain, generalize to unseen domains, or be independent to domains (Gulrajani & Lopez-Paz, 2020). However, few attempts to understand the data before using them to train the model. Similarity is a fundamental concept for understanding the relationship between different data. In deep learning, we are more concerned about similarity on semantic level. Semantically similar images can sometimes differ very much in the RGB space.

The informational similarity measure proposed by Lin (1998) measures the similarity between A and B by the ratio of the amount of information needed to describe the commonality between A and B and the amount information needed to describe A together with B:

$$sim(A, B) = \frac{I(common(A, B))}{I(description(A, B))} = \frac{K(f_B) - K(f_B|f_A)}{K(f_A, f_B)}. \tag{3}$$

It is the Jaccard similarity between A and B measured with information. A property of the informational similarity measure is universality: it does not depend on a particular representation (or modeling) of A and B. We can use model information to estimate the information terms in (3), and turn the similarity measure into (4): (using approximations $L(M_A, A) \approx K(f_A)$ and $L(M_A, B) \approx K(f_B) - K(f_B|f_A)$), see discussion on the symmetricity of $S$ in Appendix A.3)

$$S(A, B) = \frac{L(M_A, B)}{L(M_A, A) + L(M_B, B) - L(M_A, B)}. \tag{4}$$

Another benefit of information similarity is that the similarity can be measured with respect to one side, resulting in a unidirectional similarity measure:

$$S^{uni}(A, B) = \frac{I(common(A, B))}{I(description(B))} = \frac{L(M_A, B)}{L(M_B, B)}. \tag{5}$$

Note that generally $S^{uni}(A, B) \neq S^{uni}(B, A)$. The unidirectional similarity measure is useful for depicting the relationship between A and B. For example, if $S^{uni}(A, B) < 1$ and $S^{uni}(B, A) = 1$, then one can tell that A is a subset of B.

We perform experiments on two commonly-used domain adaptation datasets: Office-31 (Saenko et al., 2010) and Office-Home (Venkateswara et al., 2017). They each have 3 and 4 different domains for images of the same set of classes (Figure 4). We calculate information similarity $S$ and unidirectional information similarity $S^{uni}$ for each pair of domains in each dataset. The baseline we use for comparison is the distance of first-order and second-order statistics in the feature space

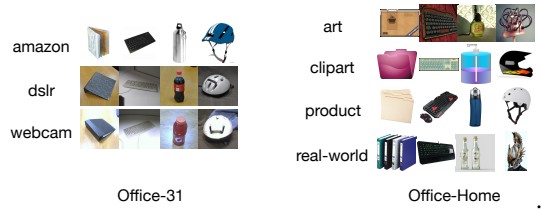

Figure 4: Image domains in Office-31 and Office-Home.

produced by ResNet-56 (He et al., 2016b) pretrained on ImageNet (Russakovsky et al., 2015) (see Appendix for details). This is inspired by the Maximum Mean Discrepancy (MMD) (Gretton et al., 2006) method for comparing data distributions. In domain adaptation literature, one often aims to minimize such distances to match domains (Pan et al., 2009).

As illustrated in Figure 5, information similarity measure give a similarity score between 0 and 1, which is intuitive and comparable across datasets. $S$ and $S^{uni}$ largely agree with feature distances, but the latter is not comparable across datasets. For instance, *dslr* and *webcam* in Office-31 are much more similar than any other pair in the two datasets, which is reflected in $S$ but not in feature distance. $S$ is also more faithful within dataset for corresponding to visual similarities. Unidirectional information similarity $S^{uni}$ gives extra information, for instance, $S^{uni}(amazon, dslr) > S^{uni}(dslr, amazon)$ shows that *amazon* contains more distinct information than *dslr*. This can be explained that each category in *amazon* has images for 90 different objects, while each category in *dslr* only has images for 5 objects.

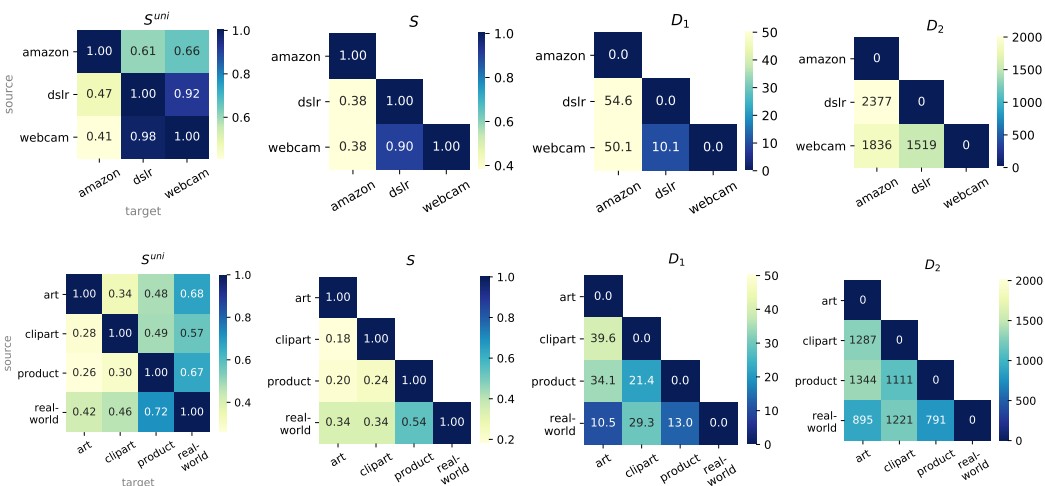

Figure 5: Pairwise domain similarity on Office-31 (top) and Office-Home (bottom). $S$ and $S^{uni}$ are information similarities, $D_1$ and $D_2$ are first-order and second-order feature distances.

## 5 MODEL STRUCTURE: CAPACITY

Neural networks are powerful learners that can scale to learn extremely large datasets given enough neurons and layers. It largely remains a mystery how to qualify the true capacity of a neural network. Capacity can be defined in different fashion, for example, Collins et al. (2017) measure capacity by the number of bits of data the network can remember, while Baldi & Vershynin (2019) define capacity as the number of distinct functions a network can represent. We can define the information capacity of a network $M$ as the maximum amount of information a model can hold for a given task $T$, directly using model information to measure:

$$C(M) = \max_{D \in T} L(M_D, D). \tag{6}$$

To illustrate the information capacity of a model, we experiment with models from small to large (4 configurations of ResNet-11: *large*, *standard*, *small*, and *tiny*. See Appendix for details), and datasets of varying complexity (subsets of TinyImageNet[1] with *50*, *100*, *150* and *200* classes.). We measure the model information of every model trained on each task, and plot results in Figure 6.

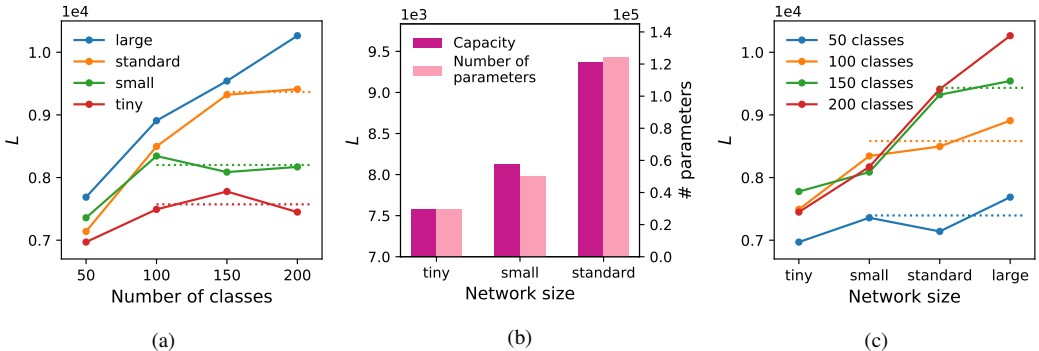

(a)            (b)            (c)

Figure 6: Measuring capacity using information. (a) the amount of information a model can store is capped by its capacity. (b) increase in capacity is correlated with an increase in the number of parameters. (c) a task has an inherent amount of information content.

The first thing we notice in Figure 6.a is that with increasing complexity of the task, model information saturates at a certain point for each model. This displays the capacity of a model (in dotted lines). Larger models can store more information and saturate later at larger tasks. Figure 6.b plots the increase of information capacity w.r.t. the increase in the number of parameters. Information capacity roughly increases with the number of parameters in the model, which agrees with the observation in Collins et al. (2017).

The same observation also apply to datasets: each dataset has a definite amount of information content. Larger models can learn more information from the dataset, but can hardly learn more than this amount. This is shown in Figure 6.c, and dataset information is indicated by the dotted lines.

## 6   MODEL PARAMETER: ABLATION

Next we turn to the analysis of parameters within a network. A common way to understand components in a network is to perform ablations: remove some units or structural components from the network, and compare the performance before and after ablation (Meyes et al., 2019). A larger performance drop signifies larger importance of the ablated component for performing on the task.

However, there are several caveats to this approach: firstly, ablations cannot reveal the true contribution of a component "*in vivo*." If a network layer is removed and the model re-trained, the functionalities of the layer can get substituted by other layers. In this case, ablations fail to reveal the contribution of the layer in the original network. Secondly, ablation cannot be performed on components vital to model performance, for example, if residue connections are removed from a very deep CNN, the network can fail to train. It would be unreasonable to therefore conclude that residue connections contribute $100\%$ to the model.

Here we propose to use model information to measure the contribution of a network component. The information $I$ in a network component $c$ can be measure in the following fashion (where $M^{\bar{c}}$):

$$I(c) = L(M_D, D) - L(M_D^{\bar{c}}, D). \tag{7}$$

To calculate the information in $c$, we reset all parameters within this component to their initialized value before training. Then we measure the model information of this ablated model $M_D^{\bar{c}}$. The difference of information in $M_D$ and $M_D^{\bar{c}}$ is the information content in the parameters of component $c$. In effect, we are performing *information ablation*: only information stored in parameters is ablated, the model structure is kept intact.

---

[1]https://tiny-imagenet.herokuapp.com

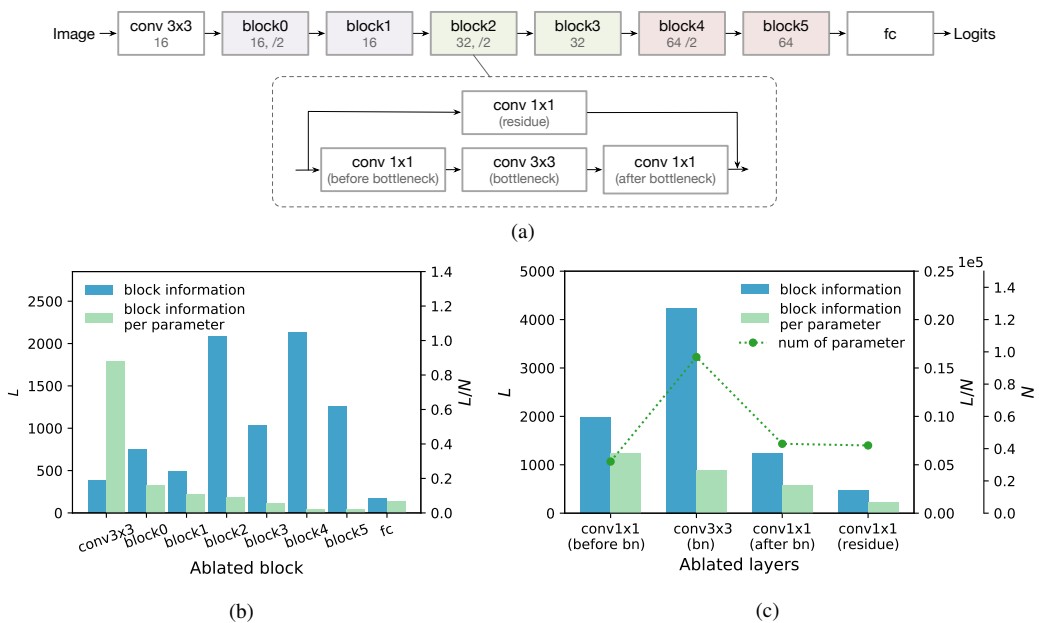

Figure 7: Information ablation. (a) Structure of a variant of ResNet model in (He et al., 2016a). (b) Information ablation by each block in the model. (c) Information ablation by different types of convolution layers. 'bn' stands for the bottleneck layer.

Information ablation helps us uncover how the information in each component contributes to the whole network, or in other words, how information in a network is stored by its components. In Figure 7, we perform information ablation on a ResNet model as an example. The model consists of a series of residue blocks, and each block contains several convolutional layers. Figure 7.b shows the information in each block, as well as the information per model parameter ("density"). There are several observations: blocks in the middle (block2 - 5) contribute the most information, blocks near input have larger information density, and blocks that reduce spatial resolution (with '/2' in notation) contains more information than blocks that preserves resolution. The last phenomenon is likely because convolution layers in downsampling blocks do more job of combining smaller object parts into larger parts, thus having more knowledge about the constitution of objects.

Figure 7(c) illustrates something unique to the information ablation method: this time we measure information contribution by four different kinds of convolution layers in each and every block. We found out that the majority of information resides in the 3x3 bottleneck layer, which is where most spatial feature transformations take place. Convolution layers on the residue path contain surprisingly little information, despite having roughly the same number of parameters as the layers before and after the bottleneck. This indicates that residue layers serve more of a structural functionality to ease training, rather than doing substantial information processing.

## 7    ALGORITHM: DISTILLATION

Knowledge distillation is a method to transfer knowledge from larger teacher models to smaller student models, which can result in better performing compact models (Hinton et al., 2015). Student models are trained with output probabilities of a teacher model in addition to the true labels. It is speculated that the student model can benefit from "dark knowledge" in teacher's predictions. If "dark knowledge" means the presence of extra information, with proper tools, we can verify the existence of extra information provided by the teacher model and quantify how much information is transferred in distillation.

We could measure the information transferred from teacher to student by the following formula:

$$I = L(M_{D+D_T}, D) - L(M_D, D). \tag{8}$$

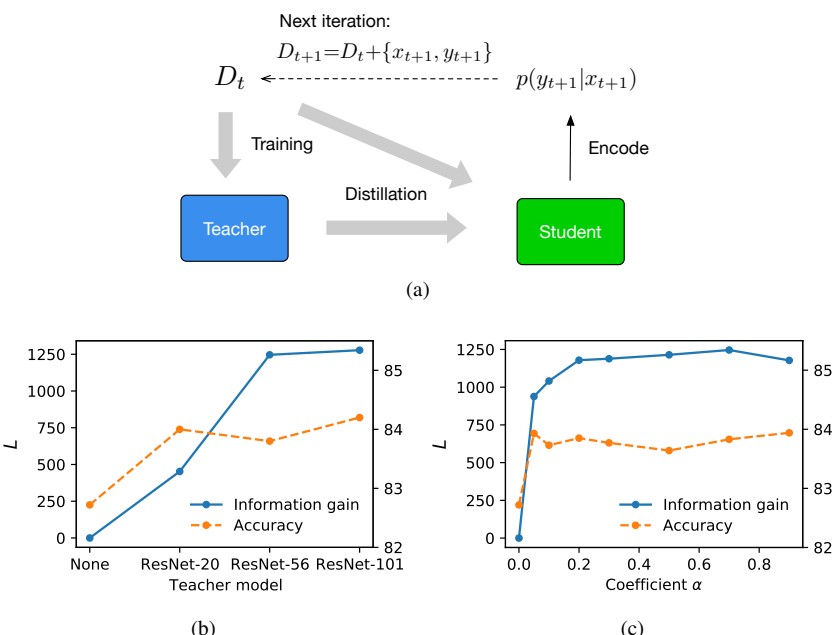

Figure 8: (a) illustration of the training process in synced distillation. ➡ indicates training procedures with information flow on the arrow directions. (b) information gain of students from different teachers. (c) information gain of the student when trained with different distillation coefficient $\alpha$.

$D_T$ is teacher's predictions on dataset $D$. However $L(M_{D+D_T}, D)$ cannot be directly measured with online coding because $D_T$ already contains much information about the labels, and the student model having access to $D_T$ can encode $D$ with minimal effort by merely looking at $D_T$. We propose a method called *synced distillation* to calculate online codelength of dataset $D$ with the presence of a teacher model.

The process of online coding gradually increases the size of the dataset $D$ and updates the model accordingly. In synced distillation, we update the teacher and the student model in synchronization (Figure 8.a): in each iteration, we train the teacher with a subset $D_t$, then using $D_t$ and teacher's predictions to train the student via distillation. The student model is then used to encode $y_{t+1}$. In the end, the student will be the same as training with conventional distillation. But during the process, the teacher never leaks any information about future examples to the student. This enables the student to generate a valid encoding of $D$, which can then be used to calculate model information.

As pointed out in Zhang et al. (2020), model information can help illustrate a different aspect of learning dynamics than model performance. For the simple 5-layer CNN model that we use as student, distillation with different teachers and distillation coefficient $\alpha$ yields similar performance (Figure 8.b and c). However, from measuring the information gain $I$, ResNet-56 and ResNet-101 transfer more information to the student than ResNet-20. This helps the student reach lower cross-entropy with fewer examples, although the final performance is similar. A larger distillation coefficient also increases information transfer. This shows that as the weighting of the KL-divergence term increase in the student's loss function, the student learns more from the teacher. Observations such as this can help analyze and design better algorithms or understand why algorithms like knowledge distillation lead to performance gain (Yim et al., 2017).

## 8 DISCUSSION

Currently, methods for estimating the description length of neural network models are still quite preliminary, lacking through analysis and guarantees of their efficiency. Using prequential coding to estimate model information or Kolmogorov complexity also introduces dependency on model architecture and training procedure, for example, dropout, batch normalization, and SGD optimizer will all affect the model information estimations. Because prequential codelength is always larger

than Kolmogorov complexity, we can optimize the training hyperparameters to achieve as lower codelength as possible, which makes the estimation tighter.

When applying model information to perform analysis, as we did in this paper, it is also necessary to vary only one variable at a time. Because $L(M, D)$ is a function of both the model and the dataset, when analyzing models, the dataset needs to be fixed for codelengths to be comparable. Similarly when datasets are the center of interest the model needs to be fixed. This also introduces dependency. For instance, theoretically we can define difficulty by $K(f)$, but empirically when estimating $K(f)$ using $L(M_D, D)$, dependency on model architecture becomes inevitable. One can choose any adequate model architecture for training on the task, and fix that to compare the difficulty of tasks.

Nonetheless, we demonstrate that model information could be a powerful analysis tool in deep learning. To summarize, the key observations from our experiments are:

- Model information allows us to quantify properties such as difficulty, capacity, and similarity, especially for complex data and deep models, in a consistent fashion.
- Model information can help analyze and understand learning in deep networks by showing how information is transferred and stored.
- Such a tool is widely applicable to many kinds of problems because it does not depend on specific models or learning algorithms.

We hope problems discussed in this work serve as examples of the versatility of an informational perspective in investigating neural network learning.

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

## A    Experiments

### A.1    General experiment settings

For experiments in this paper, we use (2) to estimate model information and set the encoding set size $k$=10000 (except for experiments on Office-31 and Office-Home, see below). All models are trained using Adam optimizer and early-stopping on the validation set.

### A.2    Difficulty

Statistics of noise injection experiments are given in Table 2-4. We trained ResNet-56 models on MNIST with different kinds of injected noise. Model confidence is defined as the margin between the probability of the top and the second class predicted by the model:

$$confidence = \mathbb{E}_x[p_{c_1}(x) - p_{c_2}(x)] \tag{9}$$

$$c_1 = \arg\max_k p_k(x) \tag{10}$$

$$c_2 = \arg\max_{k \neq c_1} p_k(x) \tag{11}$$

Table 2: Model statistics on MNIST with image noise (missing pixels)

| Noise level | 0 | 0.05 | 0.1 | 0.2 |
|---|---|---|---|---|
| $L(M_D, D)$ | 1559 | 1718 | 2120 | 2719 |
| Accuracy | 0.996 | 0.992 | 0.988 | 0.981 |
| Confidence | 0.994 | 0.989 | 0.985 | 0.985 |

Table 3: Model statistics on MNIST with image noise (missing objects)

| Noise level | 0 | 0.05 | 0.1 | 0.2 |
|---|---|---|---|---|
| $L(M_D, D)$ | 1559 | 1489 | 1560 | 1296 |
| Accuracy | 0.996 | 0.953 | 0.909 | 0.865 |
| Confidence | 0.994 | 0.946 | 0.899 | 0.848 |

Table 4: Model statistics on MNIST with label noise (massart noise)

| Noise level | 0 | 0.05 | 0.1 | 0.2 |
|---|---|---|---|---|
| $L(M_D, D)$ | 1559 | 1833 | 1735 | 1605 |
| Accuracy | 0.996 | 0.949 | 0.908 | 0.867 |
| Confidence | 0.994 | 0.973 | 0.941 | 0.908 |

### A.3    Domain similarity

We use ResNet-18 model pre-trained on ImageNet as the initial model, then finetune on source dataset $S$ and measure $L(M_S, T)$ on target dataset $T$. Pairwise codelength representing $L(M_T, T) - L(M_S, T)$ are given in Table 5-6. Diagonal values are $L(M_T, T)$.

$L(M_T, T) - L(M_S, T)$ can be directly calculated using (2):

$$L(M_T, T) - L(M_S, T) = L_{M_S}^{\text{preq}}(y_{1:k}|x_{1:k}) - L_{M_T}^{\text{preq}}(y_{1:k}|x_{1:k}) \tag{12}$$

This assumes that information from ImageNet is known and is not included in calculating similarities. It is possible to use random models as initialization, but as the number of examples is too small for some domains in Office-31, we found that using pre-trained models gives more stable estimation of codelength.

Also, because the smallest domain in Office-31 and Office-Home only have 399 and 1942 training examples respectively, we use about half of the dataset, i.e., $k$=200 for Office-31 and $k$=1000 for Office-Home to estimate $L$ in (2).

Theoretically, the similarity measure in (3) and (4) is symmetric:

$$S(A, B) = \frac{I(common(A, B))}{I(description(A, B))} = \frac{K(f_B) - K(f_B|f_A)}{K(f_A, f_B)} = \frac{K(f_A) - K(f_A|f_B)}{K(f_A, f_B)} = S(B, A)$$

which is because $K(f_A) + K(f_B|f_A) = K(f_B) + K(f_A|f_B) = K(f_A, f_B)$. Empirically, when using $L(M_A, B)$ as an approximation of $K(f_B) - K(f_B|f_A)$, usually $S(A, B) \neq S(B, A)$ because $L(M_A, B)$ and $L(M_B, A)$ doe not equal exactly. In this paper we report empirical results using (4).

Table 5: $L(M_T, T)$ - $L(M_S, T)$ on domains in Office-31

| Source \ Target | amazon | dslr | webcam |
|---|---|---|---|
| amazon | 263.0 | 124.7 | 106.3 |
| dslr | 139.4 | 318.4 | 26.6 |
| webcam | 153.9 | 6.6 | 316.6 |

Table 6: $L(M_T, T)$ - $L(M_S, T)$ on domains in Office-Home

| Source \ Target | art | clipart | product | real_world |
|---|---|---|---|---|
| art | 2776.4 | 1076.2 | 764.8 | 451.9 |
| clipart | 2001.2 | 1630.3 | 756.8 | 617.5 |
| product | 2047.6 | 1144.8 | 1480.2 | 466.0 |
| real_world | 1609.2 | 879.2 | 411.9 | 1420.2 |

The first-order and second-order feature distance $d_1$ and $d_2$ are defined as:

$$d_1 = ||\mathbb{E}_x[f_A(x)] - \mathbb{E}_x[f_B(x)]||_2 \tag{13}$$
$$d_2 = ||Cov_x[f_A(x)] - Cov_x[f_B(x)]||_F \tag{14}$$

where $f_A(x)$ and $f_B(x)$ are the representations of $x$ produced by model trained on domain $A$ and $B$. $F$ stands for Frobenius norm.

## A.4 CAPACITY

To show the capacity of a model, we need to saturate a model by information in the training set. Therefore we choose to use ResNet-11, the smallest of ResNet configurations, and use four different layer width in ResNet-11, as follows: Large: [16, 48, 96], Standard: [16, 32, 64], Small: [16, 24, 32], Tiny: [16, 16, 24]. For example, [16, 48, 96] means the layer width is 16, 48, and 96 in the first, second, and the third residue block. The datasets we use are subsets of Tiny-ImageNet, each containing 50,100,150, and 200 classes. The number of examples in each dataset is fixed at 12500.

$L(M_D, D)$ for models of different size, and datasets of different complexity, are listed in Table 7.

Table 7: $L(M_D, D)$ for model and dataset of different size

| Dataset \ Model size | Large | Standard | Small | Tiny |
|---|---|---|---|---|
| 50 class | 7687.0 | 7140.0 | 7358.2 | 6970.1 |
| 100 class | 8907.8 | 8496.2 | 8343.9 | 7492.2 |
| 150 class | 9540.7 | 9322.1 | 8086.6 | 7777.0 |
| 200 class | 10264.0 | 9410.7 | 8170.3 | 7448.8 |
| Number of parameters | 264657 | 124433 | 49905 | 29457 |

## A.5 ABLATION

We train a ResNet-20 model (referred to as $M_D$) on CIFAR-10, reset the parameters of some network component $c$ (this ablated model is referred to as $M_D^{\bar{c}}$) and measure the difference of model information by:

$$I(c) = L(M_D, D) - L(M_D^{\bar{c}}, D) \tag{15}$$
$$= L_{M_D^{\bar{c}}}^{\text{preq}}(y_{1:k}|x_{1:k}) - L_{M_D}^{\text{preq}}(y_{1:k}|x_{1:k}) \tag{16}$$

Ablation results for different network layers are shown in Table 8-9.

Table 8: Information $I$ in different kinds of convolutional layers

| Model component $c$ | $I(c)$ | # param in $c$ | $I$ per param |
|---|---|---|---|
| conv1x1 (before bottleneck) | 1985 | 32000 | 0.062 |
| conv3x3 (bottleneck) | 4243 | 96768 | 0.044 |
| conv1x1 (after bottleneck) | 1231 | 43008 | 0.029 |
| conv1x1 (residue) | 467 | 41984 | 0.011 |

Table 9: Information $I$ in different residue blocks

| Model component $c$ | $I(c)$ | # param in $c$ | $I$ per param |
|---|---|---|---|
| conv3x3 | 379 | 432 | 0.877 |
| block0 | 751 | 4704 | 0.160 |
| block1 | 485 | 4544 | 0.107 |
| block2 | 2088 | 23808 | 0.088 |
| block3 | 1030 | 17792 | 0.058 |
| block4 | 2137 | 94720 | 0.023 |
| block5 | 1256 | 70400 | 0.018 |
| fc | 167 | 2570 | 0.065 |
| all | 5900 | 219482 | 0.027 |

## A.6 DISTILLATION

In distillation experiments, we use LeNet-5 as student and ResNet-20, ResNet-56, and ResNet-101 as teachers. The loss when training student model is given in (17).

$$\mathcal{L} = \alpha\mathcal{L}_{KD} + (1 - \alpha)\mathcal{L}_{ce} \tag{17}$$
$$\mathcal{L}_{KD} = -T^2 \sum_k p_k^t(x) \log p_k^s(x) \tag{18}$$
$$p_k(x) = \frac{e^{p_k(x)/T}}{\sum_j e^{p_j(x)/T}} \tag{19}$$

We also experiment with different coefficient $\alpha$, to control the influence of teacher models. Results are shown in Table 10-11. We observe that the student's information gain plateaus after $\alpha = 0.2$, which kind of agrees with the conventional choose of $\alpha = 0.1$ in knowledge distillation.

Table 10: Distillation experiments with different teacher model

| Teacher | Teacher accuracy | Student accuracy | $I$ |
|---|---|---|---|
| None | - | 82.7 | 0.0 |
| ResNet-20 | 87.07 | 84.0 | 452.6 |
| ResNet-56 | 88.78 | 83.8 | 1246.2 |
| ResNet-101 | 89.69 | 84.2 | 1277.7 |

Table 11: Distillation experiments with different coefficient $\alpha$

| $\alpha$ | $T$ | Accuracy | $I$ |
|---|---|---|---|
| 0 | 4 | 82.72 | 0.0 |
| 0.05 | 4 | 83.93 | 938.1 |
| 0.1 | 4 | 83.73 | 1040.9 |
| 0.2 | 4 | 83.85 | 1178.9 |
| 0.3 | 4 | 83.77 | 1188.6 |
| 0.5 | 4 | 83.64 | 1214.4 |
| 0.7 | 4 | 83.83 | 1246.2 |
| 0.9 | 4 | 83.94 | 1177.5 |

