# OpenReview forum: "Model information as an analysis tool in deep learning"
_ICLR.cc/2021/Conference — Reject_

### Official Review · AnonReviewer4 · 2020-10-27
**Original approach requiring more rigorous foundation and justification**

**Rating:** 4
**Confidence:** 3

**Review:**


This paper discusses the use of model information to assess properties of deep learning models and problems. In particular, it illustrates how model information may be used to capture the difficulty of a task, the degree of similarity between domains, the capacity of a model

The idea of using a quantity related to codelength to assess properties of a learning model is interesting; indeed quantifying the properties chosen by the authors (task complexity, domain similarity) would be certainly useful. However, I am dubious whether the assessment using model information would be reliable.

To my understanding model information evaluates the amount of information a trained model M_D contains about dataset D as a difference in codelength of the data and codelenghth of model+data (as explained in Section 2). Critically, it seems to me that this quantity depends both on the data AND the model. It is therefore a particular measure for a specific model (or at most a family of models) and a dataset. (Incidentally, although guessable, it would be proper to define all the symbols in the equations).

In Section 3, it is not clear to me how this measure may be used to evaluate, for instance, the difficulty of a task: how is M_D chosen? Is there an underlying claim that this measure would be independent from the choice of model? What is the sensitivity of information measure to the choice of the model? I am also perplexed by the results of the simulations: it seems that dropping samples or scrambling labels makes the task easier; this, though, is quite counterintuitive as it would suggest that to simplify the problem we could literally drop samples or scramble labels (I guess, but I may be wrong, that this outcome is due to the fact that "difficulty" is not evaluated in terms of generalization).

Similarly, in Section 4, it seems to me that a discussion on M_A and M_B is lacking. It is stated that a property of informational similarity is its independence from particular representations/modelling. This seems to me not to hold for model information where we have to rely on M_A and M_B.

In Section 5, how is a given task T defined? This particularly important as it defines the set over which the maximization is computed and with respect to which capacity is defined. My intuition is that computing this quantity on a small set of tasks would return at best a lower bound.

I have some doubts also in Section 7, on Equation (8). It would seem to that the model information of the student should be relative to a different dataset, one enriched with the output probabilities generated by the teacher. Why is it not?

---

> ### Author Response · Authors · 2020-11-23
> **Response to Reviewer4, about the dependency of the proposed metrics on dataset and model, and other questions**
>
> Thanks for your comments! Dependency on dataset and model is a practical limitation of our proposed metrics, which we will explain below.
>
> Model information depends both on the data AND the model: in our definition, model information $L$ is a function of both the dataset and the model. One might argue that in the absence of theoretical bounds, $L$ is less comparable when both the dataset and the model is different. However, in most cases we are concerned with only one variable, either the dataset or the model. In most experiments of this paper, when comparing codelength values, we only change one of the parameters within task, data, model, and algorithm. (An exception is Section 5, when studying capacity, we let both model and dataset change. But models are within a family of ResNet-11, and datasets are also different subsets of the same Tiny-ImageNet dataset.) And results show that, when allowing only one parameter to vary at a time, model information does behave intuitively.
>
> We add discussion on the dependency of the proposed metrics on data and model in Section 8. We also discussed how one can use model information to perform useful analysis despite these inevitable dependencies.
>
> Difficulty: similar to the above question, although theoretically we can define difficulty by $K(f)$, empirically when estimating $K(f)$ using $L(M_D,D)$, dependency on model architecture becomes inevitable. To compare the difficulty of tasks one would need to use a fixed model architecture. For any reasonable choice of the model, the trend of model information on different tasks should be similar. Here different model architecture represents different prior information $p$, and difficulty is measured relative to this prior information (i.e., $K(f|p)$). In Figure 3, missing objects and massart noise do not change L by much, the fluctuations we believe is due to the variation of the prequential codelength.
>
> Domain similarity: here $M_A$ and $M_B$ need to be the same model architecture. In theory, similarity can be defined without referring to specific modeling, but empirically we can only use methods such as prequential coding to approximate the information quantities. Therefore the result would depend on how models are chosen and trained, but as long as these are fixed, we can make comparisons between different datasets.
>
> Capacity: to evaluate the capacity of a model, the task T should be complex enough to saturate the information capacity of the model (i.e., when further increasing the complexity of the task, L no longer increases).
>
> Equation 8 in distillation: by $L(M_{D+D_{T}}, D)$, we measure the student model’s information about dataset $D$. By $L(M_{D+D_{T}}, D+D_T)$, we can measure the model’s information about both the dataset $D$ and the output of the teacher model. The information transferred from the teacher to the student is partially about the dataset and partially about the teacher. In this section we mainly focused on the former, to show that distillation increased the student model’s knowledge about the dataset. When the latter is of concern, we can use $L(M_{D+D_{T}}, D+D_T)$ to calculate.

---

### Official Review · AnonReviewer3 · 2020-10-28
**Good analysis paper, but a few concerns and questions**

**Rating:** 6
**Confidence:** 3

**Review:**

The central concept of this paper is model information, a description length of a discriminative model. The authors advocate usage of model information for analyzing several aspects in deep learning. In particular, they show how model information can be used to judge about difficulty of supervised tasks, domain similarity, model capacity, roles of different network components, and knowledge distillation. All of these are important topics and are relevant to the ICLR community. While most of the definitions and interpretations seem intuitive and valid, there are a few concerns and questions, and in some cases, it is hard to decide whether the conclusions of the interpretations should be trusted.

**Model information**
- The term *model information* is vague and may create associations with other quantities like the Shannon mutual information model has about the training dataset, $I(\theta : D)$. I suggest to clarify early that by model information it is meant some kind of description length of the discriminative model $p_\theta(Y \mid X)$.
- As model information is not a well-defined concept, it is not appropriate to say "an approximation of model information". Better to say "an instance/a variant/a definition of model information".
- As all experiments are done using the instance of model information defined by Zhang et al. [1] (called Information Transfer), the conclusions may not hold for other instances of model information. For example, if one defines model information with the prequential coding of Bleir and Ollivier [2], then increasing the number of examples with noisy labels will make the task more difficult.
- Please clarify whether the $k$ examples in equation (2) are from the training set $D$ for not.

**Task difficulty**
I really like using model information for assessing task difficulty. The only concern is that the task difficulty depends on the network and the training algorithm. Additionally, I suggest to plot task difficulty versus the number of training examples. Will we see that it plateaus after enough number of training examples are given? How would task difficulty behave in the small data regime?

**Domain similarity**
- What do union and intersection signs mean for datasets in equation (4)? If they mean the same as in equation (3), you can just repeat the same notation.
- Please explain how the right-hand-side of equation (4) fits into the framework of equation (3).
- When reading that $S^\text{uni}(A, B)$ is asymmetric, one might imply $S(A,B)$ is symmetric, which is not true. Please clarify that both measures are unidirectional.
- "For example, if $S^\text{uni}(A, B) < 1$ and $S^\text{uni}(B, A) = 1$, then one can tell that A is a subset of B." Please explain why is this sentence true?
- The definitions of domain similarity of equations (4) and (5) seem a little bit arbitrary. Why should they be defined like that? How do these domain similarity measures compare to other measures, such as domain similarity computed by Task2Vec [3].
- For easier interpretation of Fig. 5, you can use the same color map in all subplots.

**Model capacity**
- As I understand model capacity defined in equation (6) depends on the task. Why shouldn't one take supremum over tasks too?
- In the experiments of Fig. 6, do all datasets have the same size?
- The straight lines in Fig. 6c don't approximate the curves well. To have more points on the horizontal axis you can consider Resnet-[18/35/50]-k networks and vary $k$.

**Ablation**
- How do we know that quantity defined in equation (7) truly captures how important a component is and that we should trust the conclusions of experiments presented in Fig. 7?

**Minor notes**
- To make the paper more self-contained, please mention in the main text which definition of model information is used in the experiments. The same applies for model confidence.
- In eq. (7), $M_D^{\bar{c}}$ should be defined beforehand.

*References*
[````1] Xiao Zhang, Xingjian Li, Dejing Dou, and Ji Wu. Measuring information transfer in neural networks. arXiv preprint arXiv:2009.07624, 2020.
[2] Leonard Blier and Yann Ollivier. The description length of deep learning models. NeurIPS 2018.
[3] Achille, Alessandro, et al. "Task2vec: Task embedding for meta-learning." Proceedings of the IEEE International Conference on Computer Vision. 2019.

---

> ### Author Response · Authors · 2020-11-23
> **Response to Reviewer3, regarding definitions used in the paper, and a few other questions**
>
> Thanks for your comments! These are very valuable suggestions that help us clarify important details in the paper.
>
> Definition of model information: the definition of model information in the original paper is indeed not rooted in a concrete theoretical quantity, and may create confusion with notions like mutual information. Upon communication with [1], we realize that model information corresponds to the Kolmogorov complexity of the model, i.e. $L(M,D)\approx K(D)-K(D|M)=K(M)-K(M|D)$. With this definition, we can say that $L$ calculated with prequential coding is an approximation of Kolmogorov complexity.
>
> We have added elaboration on the definition of model information in Section 2.
>
> Difference of using codelength in [1] and [2]: codelength in [1] calculates generalizable model information $K(M)-K(M|D)$, while codelength in [2] calculates total model information $K(M)$ (including remembered noise on the training set). In task difficulty, increasing the number of examples with noisy labels will indeed lead to higher $K(M)$, but generalizable information $K(M)-K(M|D)$ will not increase. We think defining difficulty by generalizable model information in [1] is a more reasonable choice. We have also added emphasize on "generalizable information" in the paper.
>
> In (2), the k examples used for coding is not from the training set. This is consistent with [1].
>
> Dependence of task difficulty on model architecture and training algorithm: theoretically, we can define the difficulty of a task by the Kolmogorov complexity K(f) of its input-output function Y=f(X). Because we rely on prequential coding/model information to estimate this quantity, dependency on model architecture and training algorithm seems inevitable at this point and is a limitation of our method. However, if one is only concerned with comparing the difficulty among a group of tasks, one can fix the model architecture and the training algorithm, as we did in this paper, and the results will be comparable.
>
> We also include discussion on the dependency of the proposed metrics on model architecture and training algorithm in Section 8. We also discussed how one can use model information to perform useful analysis despite these inevitable dependencies.
>
> Relationship between task difficulty and the number of examples n: if we regard the input-output relationship of a task as a distribution $Q$ over the hypothesis space $H$, then the larger the number of examples in a task, the sharper $Q$ becomes, and $K(Q)$ will also be larger. For small-data regime, basically it will be hard to distinguish between difficult tasks, but easy tasks will have a small $K(Q)$ even for small n and can be distinguished from more difficult tasks.
>
> The definition of domain similarity: thanks for pointing out the inconsistency of notations in the definition of similarity in the original submission, which we corrected in this version. We also add equivalent representation with Kolmogorov complexity. In Equation (4), we use $L(M_A, B)$ as the common information in A and B. $ L(M_A, A) + L(M_B, B) - L(M_A, B)$ is the sum of information in A and B, minus their common information, which is the information needed to describe A and B together.  These can be seen from the connection between $L(M, D)$ and $K(f)$.
>
> Symmetry of $S(A,B)$ and $L(M_A, B)$: theoretically speaking, if model $M_A$ contain all information about dataset $A$, i.e., $K(A|M_A)=0$, and $M_B$ contains all information about dataset $B$, then based on our definition, $L(M_A, B) = K(B)-K(B|M_A) = K(B)-K(B|A)=K(A)-K(A|B)= K(A)-K(A|M_B)= L(M_B, A)$, where K is Kolmogorov complexity and $K(A,B) = K(B)+K(A|B)= K(A)+K(B|A)$. In practice, because $L$ is estimated with prequential coding, it is very likely that $L(M_A, B) \ne L(M_B, A)$. As in this paper we focus on empirical results, we assume $L(M_A, B) $ and $L(M_B, A)$ do not differ by much and we report results using $L(M_A, B)$. So theoretically $S(A,B)$ is symmetric, while empirically it is usually not.
>
> We have added discussion on the symmetric property of $S(A,B)$ in Appendix A.3
>
> The subset claim: $S^{uni}(A,B)<1$ and $S^{uni}(B,A)=1$ means $L(M_A, B) < L(M_B, B)$ and $L(M_B, A) = L(M_A, A)$, which in turn means $K(B|A)>0$ and $ K(A|B)=0$ under the same condition as in the above point. This states that once we know B, we also know A, but only knowing A does not completely know B. This happens when A is a subset of B.

---

> > ### Author Response · Authors · 2020-11-23
> > **(continued)**
> >
> > Comparison with Task2Vec: thanks for pointing out these related works on task similarity. Task2Vec is based on Fisher information in the space of output distributions, which is different from our used Kolmogorov complexity in the hypothesis space. Our method is more general in that it does not require a shared backbone network as a feature extractor, therefore can be used in tasks that do not have shared feature extractors. In our current paper, our main aim is to use empirical study to show the potential use of model information as a widely-applicable tool, therefore we focused on the versatility of applications and did not perform thorough comparison with existing approaches in each application. We believe there is much potential for future work that provide more throughout evaluation of the performance of the metrics in these applications.
> >
> > Model capacity: to evaluate the capacity of a model, the task T should be complex enough to saturate the information capacity of the model (i.e., when further increasing the complexity of the task, L no longer increases). Maximization over all possible tasks makes the definition independent of tasks, but also makes it intractable to calculate in practice. To present empirical results in this paper, we restrict ourselves to a given family of tasks.
> >
> > In experiments in Figure 6, all datasets have a fixed size of 12500 examples (Tiny-ImageNet has 25000 examples for 50 classes, which is divided by 2 so that training set and coding set do not overlap. For 100, 150 and 200 classes we keep using this number). Because model information depends on dataset size, to make comparisons the dataset size needs to be fixed.
> >
> > Ablation: what we try to propose is an alternative ablation method that components conventional ablation. Because importance of a component can be defined differently, our parameter ablation method corresponds to a definition of importance by the amount of information a component carries. One evidence that could suggest the values in Figure 7 are reasonable, is: if we add up the information in each layer of the network, we get 141% of the information of the whole network ($L(M_D, D)$). And if we add up by the type of convolutional layers, we get 134% of the total information. Because there is very likely to have some degree of duplication of information in adjacent layers, this 130-140% number seems to suggest the individual information values are quite reasonable (at least when comparing to the information of the whole network).
> >
> > Other issues on the writing and presentation of the paper: we have updated the paper to incorporate these helpful suggestions.  We also added some of the above discussed details to the paper.

---

### Official Review · AnonReviewer2 · 2020-10-28
**Promising paper of case-studies for model information. Reservations about rigour and superficial treatment arise, however.**

**Rating:** 4
**Confidence:** 2

**Review:**

The paper examines different use-cases of a quantity proposed in prior works which is said to capture the model information. It shows that this quantity behaves as expected overall. Quantifying the amount of information a deep neural network is a very interesting question for the community with both theoretical and practical appeal (from an analytical perspective but also from a model selection perspective, for example).

Overall, I’m scoring the paper with a reject. While the model information seemingly behaves as expected, the weaknesses in the exposition, as well as potential mistakes and a lack of rigour in the paper, will require careful editing.

Quantifying model information can be an important tool. This paper sets out to show that quantities defined by Voita & Titov (2020) and Zhang et al (2020), respectively, seem meaningful and behave as one would expect.

However, the examples lack falsification and depth.

The structure of the ResNet in 7(a) is either wrong or non-standard: The residual connection is usually just an identity function: `out = in + block(in)`. Is this intentional? And if it is, what does this tell us about regular ResNets then?

Similarly, for the model ablation, it is not clear why resetting the parameters to their initial parameters is the appropriate method of ablation.

Another potential issue is that the definitions in Section 4 pretend that $L(M_A, B) = L(M_B, A)$ when it is clearly not. See also [Xu 2020](https://arxiv.org/abs/2002.10689) for example.

This reviewer would wish for a more in-depth treatment of the various examples in order to be thoroughly convinced.

### Minor concerns

Figure 1 uses a line plot, even though the x-axis refers to unordered categories.

---
### Rebuttal

I thank the authors for their detailed reply. I still consider the contribution to be too high-level and to cover too much ground without going into sufficient depth. I am not sure I can follow the argument about Kolmogorov complexity. Its chain rule is also only equal up to a logarithmic factor. I will keep my score the same.

---

> ### Author Response · Authors · 2020-11-22
> **Response to Reviewer2, about the aim of the paper, some details in experiment settings**
>
> Thanks for your comments! We will explain below regarding the concerns:
>
> Falsification and depth in examples: in this paper, we do not aim to provide state-of-the-art solutions to each of the problems in Section 3-7. This paper aims to use empirical study to show the potential use of model information as a widely-applicable tool, and verify that model information can lead to new insights or new attacks in deep learning problems. Therefore, we did not go into great length into the theoretical details of each method or thorough comparison with existing approaches. To show that model information is fairly universal, we focus on the width of applications by experimenting with different problems in multiple areas. We believe there is much potential for future work that addresses each of these problems more rigorously.
>
> That said, there is indeed some lack of explanation and inconsistencies in the theoretical details of this paper. We have improved some details in the revision. For example, we add more discussion about the definition of model information L(M,D) in Section 2, and we change to use better notations in the definition of similarity in Section 4 and Appendix A.3.
>
> ResNet architecture: The ResNet architecture we use is from [1], an improved version of the original ResNet [2]. In [1], there is a convolutional layer on the residue path whenever the input and output dimensions of the residue block are not the same ([3]). Similarly, in [2] there is a linear layer on the residue path (Equation 2 of [2]). Therefore, Figure 7.a represents a general picture of where the parameters are in a typical ResNet model.
>
> Parameter ablation: we do not intend to propose a better ablation method, but an alternative ablation method that components conventional ablation. It can provide different insight about the contribution of model components, and can be used when conventional ablation is inapplicable.
> Justification of resetting parameters: resetting parameters to their random-initialized value removes the learned information about the task in the parameters. One could also set parameters to zero to remove information, but the resulting network will be difficult to optimize.
>
> Reciprocity of $L(M_A, B)$: theoretically speaking, if model $M_A$ contain all information about dataset $A$, i.e., $K(A|M_A)=0$, and $M_B$ contains all information about dataset $B$, then based on our definition (we elaborated in the revised paper, see Section 2), $L(M_A, B) = K(B)-K(B|M_A) = K(B)-K(B|A)=K(A)-K(A|B)= K(A)-K(A|M_B)= L(M_B, A)$, where K is Kolmogorov complexity and $ K(A,B) = K(B)+K(A|B)= K(A)+K(B|A)$. In practice, because $L$ is estimated with prequential coding, it is very likely that $L(M_A, B) \ne L(M_B, A)$. As in this paper we focus on empirical results, we assume $L(M_A, B)$ and $L(M_B, A)$ do not differ by much and we report results using $L(M_A, B)$.
>
> To help clarify this issue, we added discussion in Appendix A.3.
>
> Line plot in Figure 1: we agree that for unordered variables it is not quite suitable to use line plot, but in this graph we ordered the datasets from easy to difficult and use line plot to help us better show the trend of how performance changes with model information.
>
> [1] He, Kaiming, et al. Identity Mappings in Deep Residual Networks
>
> [2] He, Kaiming, et al. Deep Residual Learning for Image Recognition
>
> [3]https://github.com/KaimingHe/resnet-1k-layers/blob/fde7ee93c464d8312ca9b3df1c488cb245a9e2c1/resnet-pre-act.lua#L83

---

### Official Review · AnonReviewer1 · 2020-10-29
**An interesting survey, but being short of significant novel contributions**

**Rating:** 4
**Confidence:** 3

**Review:**

The authors propose a statistic from information theory as a tool to analyze several aspects in machine learning: data, model, etc.
use the codelength of a dataset subtracting the cross-entropy of the final model. The results are interesting, but slightly fall short of novelty, and do not lead to any significant implication/improvement to the existing machine learning pipeline/model.

Clarity&Originality&Significance:

Below I will take Section 3 as a specific example to elaborate.
In Section 3, the authors try to use L(M_D,D) to "define" task complexity, and argues that noise and task complexity are two independent directions.

It seems what the authors try to define, is simply the mutual information between X and Y, where {x_i,y_i} is drawn from the joint distribution of (X,Y). The quantity L(M_D,D) is an approximation (lower bound) of the mutual information. This is a commonly known fact, which makes the results fall short of novelty.

When the authors define the approximation as the task complexity, it becomes dependent on both model and the dataset, not task alone. Therefore, L(M_D,D) will be strongly dependent on how one trains the model, and the size of dataset. For example, with/without dropout, L(M_D,D) will differ significantly.

As a result, it is difficult to get useful take-away from the results of Section 3, such as Figure 3.

In addition, it is unclear what the implications would be if we have an approximation of mutual information between X and Y. Could we use it to improve the learning of the task?

In authors' response, I hope the authors can clarify the dependence of their metric on model, dataset, besides the task. Also, please clarify how it relates to mutual information, whether there is any related work that studies the approximation of mutual information between X and Y, and their relationship with the current work. Also, it would be interesting to know if there are any potential application for the proposed descriptive statistic on task complexity.

---

> ### Author Response · Authors · 2020-11-22
> **Response to Reviewer1, about novelty and relationship with mutual information, dependency of metric on model and dataset, and potential applications**
>
> Thanks for your comments! Please let us provide some clarification and explanation on the issues:
>
> Novelty and relationship with mutual information: model information is fundamentally different from mutual information between (X, Y), because mutual information is measured in the space of distributions over $X\times Y$, while ideally model information is measured with Kolmogorov complexity K in the hypothesis space H. Therefore they are independent quantities.
>
> Here is an example to illustrate: assume X is a random variable with high entropy, Y=f(X) and f is the identity function. The mutual information I(X;Y)=H(X) would also be high. However, the identity function f is a very simple function in the hypothesis space H, with a very small K(f). Model information measures something similar to K(f) with prequential coding, which tells how complex the function f is. A task with larger K(f) has more complex input-output relationships and is generally more difficult to learn. A task with a simple hypothesis like Y=X will generally be very easy to learn by a neural network. This is the reason we propose to define difficulty with model information.
>
> This similarly applies to other sections of the paper, for example, in Section 4 we also define quantities using Kolmogorov complexity and use model information as an approximation.
>
> To state more clearly the quantities we are studying and to help readers avoid possible confusion with mutual information, we elaborated on the definition of model information and its connection to Kolmogorov complexity in Section 2.
>
> Dependence of difficulty metric on model and dataset: in our definition, model information L is a function of both the dataset and the model. To approximate Kolmogorov complexity with prequential coding, using a specific model and dataset is inevitable in the coding process. Here using different model architecture means different prior information is introduced, and L would be different. However, for any reasonable choice of the model, the trend of model information on different tasks should be similar. To compare the difficulty of tasks by the value of L, the model will need to be fixed. Therefore task difficulty in Section 3 is measured with respect to a specific model family. The size of the dataset also affects the value of L, so it also needs to be fixed when making comparisons.
>
> We added in Section 8 a discussion around dependency on model and dataset and its implications. We also showed how one can use model information to perform useful analysis despite these inevitable dependencies.
>
> Application of task complexity: one potential use case is providing a guideline for controlling difficulty when designing tasks. Another is helping to interpret performance metrics: achieving 0.9 accuracy on a difficult task is much more significant than achieving the same accuracy on an easy task. Knowing the difficulty of tasks can also help to improve model performance, for example, in curriculum learning one can use difficulty to arrange tasks from easy to difficult to help optimization (e.g. [1]).
>
> [1] Bengio, Yoshua, et al. Curriculum learning

---

### Decision · Program_Chairs · 2021-01-07
**Final Decision**

**Decision:**

Reject

**Comment:**

This work presented a broad set of interesting applications of model information toward understanding task difficulty, domain similarity, and more. However, reviewers were concerned around the validity and rigor of the conclusions. Going into more depth in a subset of the areas presented would strengthen the paper, as would further discussions and experiments around the limitations of model information with regards to specific models and dataset sizes (as you have begun to discuss in Section 8). Additionally, reviewers found the updated paper with connections to Kolmogorov complexity interesting, but reviewers wanted a more formal treatment and analysis of the relationship.